# Effective Optical Image Assessment of Cellulose Paper Immunostrips for Blood Typing

**DOI:** 10.3390/ijms23158694

**Published:** 2022-08-04

**Authors:** Katarzyna Ratajczak, Karolina Sklodowska-Jaros, Ewelina Kalwarczyk, Jacek A. Michalski, Slawomir Jakiela, Magdalena Stobiecka

**Affiliations:** 1Department of Physics and Biophysics, Institute of Biology, Warsaw University of Life Sciences (SGGW), 159 Nowoursynowska Street, 02776 Warsaw, Poland; 2Faculty of Civil Engineering, Mechanics and Petrochemistry, Institute of Chemistry, Warsaw University of Technology, Ignacego Łukasiewicza 17, 09400 Plock, Poland

**Keywords:** immunostrips, blood typing, cellulose paper, optical detection

## Abstract

Novel high-performance biosensing devices, based on a microporous cellulose matrix, have been of great interest due to their high sensitivity, low cost, and simple operation. Herein, we report on the design and testing of portable paper-based immunostrips (IMS) for in-field blood typing in emergencies requiring blood transfusion. Cellulose fibrils of a paper membrane were functionalized with antibodies via supramolecular interactions. The formation of hydrogen bonds between IgM pentamer and cellulose fibers was corroborated using quantum mechanical calculations with a model cellulose chain and a representative amino acid sequence. In the proposed immunostrips, paper with a pore size of 3 µm dia. was used to enable functionalization of its channels with antibody molecules while blocking the red blood cells (RBC) from channel entering. Under the optimized test conditions, all blood types of AB0 and Rh system could be determined by naked eye examination, requiring only a small blood sample (3.5 µL). The durability of IgM immunostrips against storing has been tested. A new method of statistical evaluation of digitized blood agglutination images, compatible with a clinical five-level system, has been proposed. Critical parameters of the agglutination process have been established to enable future development of automatic blood typing with machine vision and digital data processing.

## 1. Introduction

Biosensing devices based on functionalized microporous cellulose paper matrices have been explored as novel tools for portable diagnostics in medical emergencies and in widespread disease screening [1,2,3,4,5]. These devices offer a high performance, including fast response and high sensitivity, due to the utilization of highly efficient biorecognition principles. The paper-based biosensing devices have been recently proposed for blood type determination. In medical emergencies requiring quick blood transfusion to save lives following an accident, including in war zones and far-off areas, the blood type must be immediately known. For such situations, there is a need for an in-field point-of-care (POC) fast testing device available for blood group determination independent from highly trained personnel, specialized lab equipment, or advanced sample preparation [6,7,8].

The existing methods of blood typing include routine classical blood-typing tests such as slide and tube tests, microplate technology, column agglutination, and gel centrifugation technology [9,10,11]. In spite of the advancements in standard methods of blood typing, errors in blood group determinations for the transfusion still occur [12,13]. The modern strategies for blood typing are based on biochips [14,15], microfluidic systems [16,17,18,19], near-infrared (NIR) photon transmission [20], surface plasmon resonance images [21,22], and a quartz crystal nanogravimetric technique [23]. Other transduction methods have also been used. For instance, Chang et al. have utilized impedimetric and electroanalytical measurements for accurate determination of the degree of agglutination of red blood cells in a microfluidic system [14].

Recently, novel tools based on microporous cellulose papers have been explored for portable diagnostics in widespread disease screening, including cancer, as well as in medical emergencies, such as for in-field blood typing before transfusion [5,24,25,26,27,28,29,30,31,32]. Guan and coworkers have performed preliminary studies on the stabilization of antibodies adsorbed on paper [26]. They added various modifiers to the cellulose paper, such as dextran, polyvinylpyrrolidone (PVP), and glycerol, to enhance protection of antibodies against degradation [26]. High sensor stability could be obtained by binding antibodies to EDC-NHS activated sites in 4-carboxybenzene diazonium modified cellulose fibers [33]. Afshari et al. have utilized nitrocellulose in a blood typing kit [34].

The awareness of one’s own blood type, and the presence or absence of specific antigens on the surface of red blood cells or antibodies in plasma is an important factor in case of an urgent need for a blood transfusion. It is important especially in situations when blood from different donors is mixed. The contact with foreign antigens on the erythrocytes may cause an unwanted immune response, such as cell agglutination, which is the phenomenon of clumping of red blood cells which is associated with serious complications that may threaten the patient’s life. In order to avoid this adverse effect, a series of tests that exclude agglutination are carried out before each transfusion. Therefore, simple, fast, and inexpensive methods for blood typing are highly desired.

The microporous cellulose paper-based devices offer a high performance, including fast response and high sensitivity, due to the utilization of highly efficient biorecognition principles [1,35,36]. Moreover, they are inexpensive and do not require specialized personnel for handling. Along with excellent properties, cellulose papers also exhibit a high storage capacity of molecules in the paper microchannels. Hence, the desired reactive molecules can be uploaded into a cellulose paper strip and immobilized on cellulose fibrils in the paper microchannels to be stored in the protected dry state for later use for the detection of analytes.

According to review papers on paper-based immunostrips, the most popular substrates are Kleenex paper towel (34 g/m^2^, 140 μm thick) and Whatman paper (No. 4) [37,38,39,40]. The paper-based methodology is most sensitive to antibody dilutions between 1:1 and 1:8. Some studies have used ratios of 1:1–1:4 of blood samples with buffer or whole blood [41,42]. It should also be noted that the antibody concentration and the blood dilution depend strongly on the chosen flow method: lateral or vertical [38]. The lateral chromatographic flow method takes advantage of the fact that agglutination results in the formation of large cell aggregates, which cannot move in the paper’s porous structure. In contrast, the lateral flow blood grouping test takes advantage of the fact that agglutinated RBCs are transported a shorter distance than non-agglutinated RBCs, and the plasma is separated through the hydrophilic channels by capillary force. The resulting speed depends strongly on the density and thickness of the paper used, but detection is not always precise, especially for anti-D antibodies [38,43]. However, current paper-based diagnostics are multi-step processes and are difficult to automate, and therefore not practical for high-volume applications [38,40]. Hence the need for continuous improvement and simplification of the imaging and detection method.

In this work, we have developed a simple and fast immunostrip method for the determination of blood groups for cases of emergency requiring quick blood transfusion. The proposed assay enables protection against life-threatening blood agglutination, caused by improper blood type selection for transfusion. The blood typing with the proposed immunostrip method involves blood groups in the AB0 and Rh systems (i.e., groups: A+, A−, B+, B−, 0+, 0, AB+, and AB−). Moreover, the propensity of antibodies to form hydrogen bonds to cellulose chains was evaluated using Spartan quantum-mechanical calculations. Highlighted, the agglutination of RBCs can be clearly distinguished from a non-agglutination case using a naked eye evaluation, supported by a numerical analysis of the immunostrip images, including such factors as the number of aggregates, average aggregate size, and the area fraction obtained from an ImageJ analysis of images.

## 2. Results and Discussion

### 2.1. Principal Operation of Paper Immunostrips for Blood Typing

Human blood typing is based on the AB0 blood group system as the main classification method. This system is used to denote the presence/absence of A or B antigens on the surface of red blood cells (RBCs, erythrocytes). Thus, a person may have blood of type A, type B, type 0, or type AB. Another important main grouping of blood is the Rhesus (Rh) system which relies on the presence/absence of antigens C, c, D, e, E on the surface of RBCs. Because the most immunogenic, and thus significant, antigen of the Rh system is antigen D, the positive Rh + D (Rh+) or negative Rh-D (Rh-) blood types, with or without D protein on the RBC surface, respectively, can be distinguished. Thus, the blood group is a combination of both the AB0 and the Rh systems (e.g., AB+).

In Figure 1, the principal operation of a paper immunostrip for blood typing is depicted. The examples in panels A and B show a cellulose paper with immobilized anti-A IgM antibodies that normally exist in the plasma of blood type B but not in plasma of blood type A. The antigens present on the RBC surface are supplied with a blood droplet sample. It is shown that after the interaction of anti-A IgM antibodies with drops of two different blood types, one of two possible outcomes can be observed. Upon the addition of a droplet of type A blood, the agglutination process (positive outcome) occurs (Figure 1A). It is indicated by the formation of clumps due to the presence of antigens A on RBCs in the blood sample droplet. The agglutination process results in the formation of large aggregates of blood cells and the precipitation of the antigen–antibody complexes which are visible as the sediment on the biosensor paper surface. Upon the addition of a droplet of type B blood, the agglutination process does not occur (negative outcome) (Figure 1B). It indicates that the blood dropped contains RBCs without antigens A matching the anti-A antibodies immobilized on the surface of a cellulose paper biosensor. Therefore, the dilution of the blood and staining of the cellulose membrane is only observed. The lack of sediment on the paper indicates that the test is negative. Because the dimension of RBCs (ca. 7.2 ± 1 µm [44]) is larger than the mean pore size in cellulose membranes, the single RBCs as well as the RBC aggregates cannot pass through these membranes and must remain on the membrane surface (Figure 1C).

### 2.2. Optimization of Experimental Conditions

To achieve the best agglutination sensing performance, several important experimental parameters, such as the wettability of paper immunostrips, antibody surface coverages, and the antibody–antigen reaction time, were optimized. The aggregation of the RBCs on the paper immunostrips was evaluated using the aggregate size and the number of aggregates (counts) determined from a digital analysis of stain images using ImageJ software.

In Figure 2, the optimization of paper immunostrip wetting based on the RBC aggregate counts is presented. It is seen that the bilateral wetting of paper with PBS solution (panel (C)) results in a better immunostrip response than that observed after unilateral wetting (panel (B)). The application of standard RBC solution without prior membrane wetting with PBS shows a very weak response, visible by the naked eye (panel (A)). The formation of RBC clumps was clearly seen only when pre-wetting the paper immunostrip with PBS solution applied (Figure 2B,C). Since the bilateral wetting of the immunostrip, followed by the standard RBC solution application has led to the best expressed agglutination process of RBCs, this protocol was employed in further testing.

The size of IgM antibodies (35 ± 3 nm [45,46]) is almost 100 times smaller than the size of membrane pores (2.7–3.0 µm). Therefore, the IgM molecules can freely enter the paper pores and assemble on the cellulose fibrils’ surfaces via supramolecular interactions. After the drying of immunostrips, the antibodies remain assembled in the paper pores. Upon wetting with PBS solution, the antibodies may be released from the microchannel surface, freely diffuse to the paper surface, and interact with antigens on the surface of RBCs. Since the size of RBCs (7.2 ± 1.0 µm) is larger than that of the pores, they cannot enter the pores and the agglutination process takes place on the membrane surface, providing clear images for analysis. The time of interactions of antibodies immobilized in the immunostrips and antigens in all cases was the same. This sensitive detection by the naked eye was confirmed by the image analysis using ImageJ software (Figure 2E–G, right panels). It is seen that the number of RBC aggregates increases from 12 to 40, by switching from unilateral to bilateral membrane wetting. The percentage of aggregates with a diameter larger than 12 pixels increases from 6% for a unilateral membrane wetting to 43% for bilateral wetting (Figure 2D). Therefore, a bilateral membrane wetting with PBS solution to activate antibody modified papers was chosen for further experiments.

The effect of the time of interaction of IgM antibodies, released from the membrane, with applied standard RBC solution was investigated to determine the optimal conditions for further analyses. In Figure 2E–G, the images of aggregates formed during the blood agglutination process due to the interactions of antibodies with RBCs on the immunostrip surface (left panels) and the respective rendered images (right panels) are compared for different interaction times: 30 s, 60 s, and 90 s, in panels E, F, and G, respectively. It is seen that the percentage count of small RBC aggregates (less than 30 pixels) decreases from 97% after 30 s, to 88% after 90 s of the reaction (Figure 2H). At the same time, the percentage count of large RBC aggregates (larger than 30 pixels) increases from 3% to 12%, for the response time increasing from 30 s to 90 s (Figure 2I). Evidently, the agglutination process changes dynamically in time, and thus, the reaction time should be precisely controlled. Due to the high count of large RBC aggregates at longer interaction times, the 90 s response time was chosen for further experiments.

The key factor affecting the RBC agglutination process is the number of antibody molecules stored in membrane microchannels in relation to the number of RBCs in the blood sample. For a clear readout, the number of the assembled (agglutinated) RBCs should be large enough to form fully developed aggregates on the immunostrip surface. As a reference, we can calculate the surface coverage of RBCs in a saturated monolayer, per geometrical (flat) immunostrip surface, as:(1)ΓRBC,hor=223r2=2.23×106 RBCscm2
for horizontal orientation in a hexagonal lattice with 2 RBCs per unit cell, and:(2)ΓRBC,vert=12rh=6.17×106 RBCscm2
for vertical orientation in a square lattice, where the RBC radius is r=3.6±0.5 µm and its thickness h =2.25±0.25 µm [44]. A drop of blood sample of 3.5 µL was used in the experiments, with the average concentration of RBC of 7.88 ± 0.41 pM (taking into account differences between the blood of men and women) delivering ca. 27.6 attomols of RBCs per 1 cm^2^ of the immunostrip membrane, or ca. $1.66\times 10^{7}$ RBCs/cm^2^, which is enough to form a multilayer film or 3D aggregates.

The antibodies’ upload to the membrane channels was optimized through the analysis of immunostrip morphology after the interactions with the standard solution of red blood cells using ImageJ software (Figure 3). In Figure 3A,B, the optical and SEM images of an immunostrip surface are shown. In the experiments, the surface of 12 × 12 mm^2^ paper membranes was covered with different aliquots of antibody solutions and then a 3.5 µL sample of standard solution of RBC was added. It is clearly seen in Figure 3C, that the average size of RBC aggregates formed for antibodies’ solution volume equal to or higher than 40 µL, is almost constant and equals 27 pixels^2^. Therefore, the volume of 40 µL of an antibody solution was chosen for the immobilization of antibodies in paper microchannels. In Figure 3D, an example of a standard solution of RBCs with the immunostrip method developed is presented. It is clearly seen that the immunostrip method is capable of distinguishing agglutination and non-agglutination processes on the strip’s surface. The A−, B−, and 0+ RBCs were dropped onto IMS-44, IMS-50, IMS-542 immunostrips coated with anti-A, anti-B, and anti-D antibodies. The agglutination process was demonstrated by the formation of aggregates of RBCs during the interactions between antibodies released from membrane channels with RBC- bound antigens, respectively, for RBC A Rh- on IgM anti-A strip, for RBC B Rh- on IgM anti-B strip, and for RBC 0 Rh+ on IgM anti-D strip, using all three selected paper membranes. The negative agglutination response of immunostrips was signified by lack of sediment.

### 2.3. Molecular Dynamics Simulation of IgM Interactions with Cellulose Fibers

The paper test immunostrips are based on a cellulose matrix with its channel’s surface modified with monoclonal IgM antibodies held by supramolecular forces. The IgM molecules are pentameric ensembles of five IgG subunits held together by disulfide bonds and with a characteristic single joining (J) chain (Figure 4A). The antibody’s size of ca. 35 ± 3 nm dia., Czajkowsky and Shao, does not hinder the ability of IgMs to navigate the cellulose matrix channels since the papers for immunostrips were chosen to have an appropriately large pore size (ca. 2.7–3.0 µm) [45]. During the drying process, the antibodies interact with cellulose functional groups and form supramolecular assemblies with cellulose chains in the membrane channels. Then, during the agglutination test, they must disassociate from these supramolecular assemblies and diffuse to the membrane surface to react with RBC antigens in the blood droplet on the membrane surface. This desorption of antibodies and diffusion process is induced by cellulose membrane wetting performed before the RBCs/blood typing test.

It is seen in Figure 4B that a cellulose chain contains many -OH groups which may be involved in the formation of hydrogen bonds with amino acids of antibodies. Furthermore, the interactions of IgM antibodies with cellulose fibers have been analyzed using the molecular dynamics (MD) simulations. In Figure 4C, a cartoon depicting the interactions of IgM antibodies with cellulose fibers is presented. The MD and quantum mechanical calculations of exemplary amino acids (HIS, GLY) and a sequence of five amino acids (-SER-ILE-PHE-LEU-THR-) of the heavy chain of antibodies [47] with cellulose molecule are presented in Figure 4D,E. It is clearly shown that hydrogen bonds are formed between oxygen atom of -OH groups on cellulose moieties and hydrogen atoms of amino acids. These bonds keep the antibody molecules at a close distance to the cellulose matrix until cellulose membrane wetting begins freeing the IgMs and enabling their diffusion to the membrane surface. From the electrostatic potential maps drawn on the electron density surfaces, it is seen that the negative charge is accumulated on the oxygen atoms.

### 2.4. Morphology and Surface Structure of Cellulose Membranes

The surface morphology of pure cellulose membranes and modified immunostrips was examined using scanning electron macroscopy (SEM) images. All cellulose membranes used consisted of high-quality cotton linters containing more than 90% of cellulose [48]. In Figure 5, images of exemplary cellulose immunostrip W-50 are presented. As shown in Figure 5A, the membrane is composed of long, randomly distributed, and compact cellulose fibers. The magnified SEM image (Figure 5B) shows that the random fiber network consists of two kinds of fibers: thicker fibers with diameter ~10 µm and thinner fibers with diameter ~2 µm. The fiber’s length was in the range of 500 to 1600 µm. These results are consistent with the estimated diameter and length of microfibrillated cellulose (MFC) obtained by Bharimalla et al. [48]. The modification of cellulose papers with IgM antibodies results with no visible morphological changes. This is due to the small size of antibodies. Czajkowsky et al. have shown that human IgM complexes deposited on a mica substrate in typical cryo-AFM images are star-shaped, with a central circular region (with a diameter of 19 ± 2 nm) from which a number of arms project out radially, each 11 ± 1 nm in length and 13 ± 3 nm wide [45]. Therefore, the immobilized antibodies filled up the pores and small interfiber air spaces of cellulosic strips. The membrane permeability is determined by the fiber thickness, compactness of the fiber network (porosity), and surface properties, including hydrophobicity, surface charge, and presence of functional groups. The mean pore size in these membranes is 2.7 µm and, hence, it is smaller than the diameter of an RBC disk, which is typically 7.2 ± 1.0 µm (Figure 1). Hence, the single red blood cells, as well as the RBC aggregates, cannot pass through these membranes and must remain on the membrane surface. This means that for the agglutination to proceed, the antibodies absorbed in the membrane channels must diffuse to the membrane surface. In Figure 5C, the morphology of a cellulose paper immunostrip in absence of agglutination processes (a negative test result) is shown. It has been found that the IMS-50 membrane surface had been uniformly coated with blood and a smoothening of the cellulose surface is observed. It is also seen that the cellulose fibers are conglomerated. The inter-linking of blood with cellulose fibers are clearly discernible. The observed response in this case constitutes the blood staining of the paper. In contrast, during the agglutination process (a positive test, Figure 5D), a thick layer of RBCs is formed on the membrane surface covering all cellulose fibers and a blood clot is visible.

### 2.5. Blood Typing using Paper Immunostrips

After optimizing the immunostrip testing operation with red blood cells, the validation of paper immunostrip assays was performed using blood samples of different types (A+, B+, AB+, 0+, A−, B−, AB−, and 0−). In Figure 6, the patterns obtained in the red blood cells’ agglutination tests with antibodies for A, B, AB, and 0 blood types, with positive Rh+ (Figure 6A) and Rh- (Figure 6B) factors, on three types of immunostrips, are collected. For a successful evaluation of the test results, it is necessary to assume that the obtained agglutination patterns can be readily distinguished from uniform red spots of stained paper left by non-agglutinated RBCs. The images presented in panels A and B in Figure 6 demonstrate that the developed immunostrip assay enables a straightforward blood group determination with a naked eye evaluation. To support the naked eye evaluation, we have also developed a digital evaluation of the agglutination strength based on statistical analysis of the RBC aggregate number and their average size, using ImageJ software. In Figure 6C, the dependence of the average aggregate’s size on the antibodies surface coverage is presented. It is shown that for the agglutination process (Figure 6C, curve 1), the size of aggregates increases with surface coverage of antibodies. The mechanism of aggregation of RBCs is similar to the nucleation process. The size of aggregates for the non-agglutination process is almost on the same level independent on the surface antibody coverage, indicating that the assemblies of RBCs were not formed, and no aggregation of RBCs was occurring (Figure 6C, curve 2; see also Appendix A for the negative control). The 40 µL volume of antibody solution, dropped onto the filter paper’s surface, shows the biggest changes between aggregates after interactions with blood samples and allows to differentiate the agglutination and non-agglutination processes during blood typing using the objective factor (average aggregates size) obtained with ImageJ software.

### 2.6. Durability of Cellulose Paper Immunostrips

The stability and durability of antibodies adsorbed onto the immunostrip’s surface was evaluated by analyzing the agglutination process of RBCs after the storing of immunostrips at room temperature and at 4 °C for 1, 7, 14, 21, and 28 days. The results for exemplary immunostrips prepared using W-50 membranes and anti A, anti B, anti D antibodies are shown in Figure 7 (see also Appendix A), with color coded storage time: 1 Day (red frame), 7 Days (navy blue frame), 14 Days (green frame), 21 Days (blue frame), and 28 Days (black frame). The naked eye evaluation indicates that there is no difference in the agglutination process observed after storing immunostrips for up to 28 days (Figure 7A,B). To corroborate the results of the naked eye examination of the durability of immunostrips, we have also evaluated the agglutination process using a statistical analysis of digitized images of immunostrips. Presented in Figure 7C,D are the dependencies of the percentage area fractions of aggregates on time of immunostrip storage. The results clearly demonstrate that immunostrip storage at room temperature or at 4 °C does not influence the good responses of immunostrips for all kinds of antibodies under the study (anti-A, anti-B, anti-D). The area fraction of aggregates for immunostrips stored at room temperature remains in the range 35 ± 2% and that for immunostrips stored at 4 °C, in the range 32 ± 2%. This analysis validates the naked eye tests and confirms that the immunostrips can be used without losing the quality of all the antibodies used, even after 28 days.

To relate the results of the blood typing performed in this work to the agglutination strength levels used in clinical blood typing, it is rational to assume that under the most favorite conditions, the rate of nucleation of RBC aggregates during the blood agglutination process depends on the concentrations of antibodies and complementary antigens present on RBCs. Once the amount of antibody solution (40 µL) for a drop of blood (3.5 µL) with 7.88 ± 0.41 pM RBC concentration (the range taking into account differences between the male and female blood) is established as sufficient to enable the most efficient formation of RBC aggregates upon the agglutination process, the area coverage by aggregates on an immunostrip surface of 35.0% can be assumed as the strongest agglutination that can be observed for the test conditions, including agglutination time, wetting procedure, etc. Therefore, the five agglutination levels, from 0 to 4+ can be classified as shown in Table 1.

According to this classification, 25 out of 27 positive test results observed in images of Figure 7 indicate strong agglutination (level 4+) and two results indicate heavy agglutination (level 3+) for immunostrips stored up to 28 days. Therefore, the proposed blood typing immunostrips enable reliable, reproducible, and rapid determination of blood groups and have a high potential for application as an aid in medical emergency and for off-field conditions.

## 3. Materials and Methods

### 3.1. Materials

The immunoglobulin M-type antibodies (IgM) against red blood cells (RBCs) antigens A, B, and D, were obtained from Merck Milipore, UK (as BIOSCOT^®^, anti-A, anti-B, and anti-D monoclonal grouping reagents). The anti-A and anti-B antibodies were supplied in blue- and yellow-colored solutions, respectively, whereas the anti-D solution was colorless. Phosphate buffer saline (PBS, pH 6.9), sourced from the Regional Center of Blood Donation and Treatment (Katowice, Poland), was used as a diluent and washing solution for all antibody solutions in this study. All the reagents were stored at 4 °C.

In this paper, we present the results for 3 (the most promising) out of 13 tested Whatman cellulose filter papers. These papers: grades 44, 50, and 542, with different properties, i.e., porosity, pore diameter, and basis weight were used as the matrix for preparation of paper immunostrips IMS-44, IMS-50, and IMS-542, respectively. The papers were cut into 12 mm × 12 mm squares and used for immobilization of antibodies. The specifications of these cellulose membranes are presented in Table 2.

Blood samples of eight whole blood types (A+, A−, B+, B−, 0+, 0−, AB+, and AB−) and commercially available solutions of red blood cells (RBCs) were purchased from the Regional Center of Blood Donation and Treatment (Warsaw, Poland). Whole blood came from tested donors and were kept in standard tubes with inner walls coated with EDTA and stored at 4 °C.

### 3.2. Modification of Cellulose Paper Immunostrips with Blood Typing Reagents

The cellulose membranes (Whatman grade 44, 50, and 542), used in this work as the matrix for the preparation of blood-typing immunostrips, were modified with antibodies against antigens expressed on the RBCs surface. For this purpose, small volumes of 40 µL (unless otherwise specified) of the respective antibody solutions (Anti-A, Anti-B or Anti-D) were applied onto 12 mm × 12 mm cut cellulose paper squares and allowed to dry for 24 h at room temperature. The expression “antibody volume (µL)” used here in the graphs refers precisely to the volume of antibody that has been applied to the cellulose paper squares. The drying was carried out in a container specially prepared for these investigations. The squares were placed horizontally on the specially prepared holder with two parallel 1 mm diameter beams spaced 10 mm apart. This method of drying cellulose paper immunostrips provided the minimum contact points of papers with beams and proper air exchange for effective drying.

### 3.3. Steps in the RBCs and Blood Typing Procedure

The experiments were carried out using a 4-step procedure. The appropriate reagents were added using a micro-pipette. Free-standing membrane squares supported on two fly line beams were used in blood typing. The following steps were involved (Figure 8):  I.Immobilization of an antibody on a cellulose membrane to form the active biosensing membrane able to interact with antigens of RBCs in blood samples. A 40 µL sample of antibodies was dosed directly from the manufacturer’s reagent without dilution. Unfortunately, the absolute antibody concentration was not provided by the manufacturer. II.Beginning of analysis: wetting of a dry antibody-modified paper biosensor by sequentially dropping a PBS solution on both sides of the paper immunostrip (i.e., bilateral wetting): 6.5 µL of PBS for IMS-50 and 10 µL of PBS for IMS-44 and IMS-542 membranes.III.Application of a 3.5 µL drop of whole blood sample of known blood type (A+, B+, AB+, 0+, A, B, AB−, and 0−) dosed from the standard EDTA tube without dilution onto the top side of the paper immunostrip. The dosing procedure was the same for experiments with commercially available solutions of RBCs (concentration ~5 × 10^6^ RBCs/µL).IV.Mixing of all reagents and waiting 90 s for agglutination, followed by drying with room temperature air stream and image recording. Mixing involved a manual rotation of 90 degrees back and forth every 2 s.

The procedure described above enables a fast response of the modified cellulose paper strips toward the blood group typing. The naked eye visible evaluation of the presence or absence of agglutination of red blood cells with antibodies is made within 90 s. Additionally, the analysis of recorded images was performed using ImageJ software for statistical treatment of the agglutination processes. During the image analysis, the following characteristic parameters were determined: the number of aggregates formed, total area of aggregates, average aggregate size, and the area fraction. The procedure used for image analysis in ImageJ was: (1) converting image size to 8-bit type; (2) applying plugin which is built into ImageJ as the Process/FFT/Bandpass Filter command (large structures: 40 pixels, small structures: 1 pixels, suppress stripes: none, tolerance of direction: 5%, autoscale after filtering, saturate image when autoscaling); (3) setting the threshold (0,100); and (4) using the built-in Analyze/Analyze particles plugin (size: 0-infinity; circularity: 0-inifinity).

The molecular dynamics (MD) simulations and quantum mechanical (QM) calculations of electronic structures formed upon the interactions of amino acid molecules with cellulose were performed using density functional theory (DFT). In calculations, the Wavefunction (Irvine, CA, USA) Spartan 14 software was employed, with B3LYP functional and 6-311G* basis set. The electron density $\rho_{e}$ is expressed in atomic units, au^−3^, where 1 au = 0.52916 Å and 1 au^−3^ = 6.7491 Å^−3^.

## 4. Conclusions

We have successfully developed a reliable assay for blood typing based on microporous cellulose paper immunostrips offering a high potential for applications in medical emergencies and places far from points-of-care and a clinical environment, using a naked eye evaluation. The detection of blood groups in the AB0 and Rh systems has been achieved by modifying the paper microchannels with antibodies against A, B, or D antigens present on the RBC surface. The antibodies were stored in microchannels and bound to cellulose fibrils via supramolecular forces and hydrogen bonds. They could be readily released upon membrane wetting to interact with RBCs in a blood drop on the membrane surface. The conditions for the strongest blood agglutination have been optimized, including the way of cellulose membrane modification, wetting of membrane, surface concentration of antibodies, antigen–antibody interaction time, and others. The blood typing and immunostrip durability have been tested at room temperature and at 4 °C. The results of the naked eye examination have been corroborated with a statistical analysis of digitized images of the immunostrip surface upon blood agglutination testing. On the basis of these analyses, a five-level agglutination strength classification, compatible with clinical levels 0 through 4+, has been developed (Table 1), which aids in the reliability of the blood typing which is necessary before blood transfusion. The immunostrips developed are inexpensive and do not require specialized personnel for handling.

## Figures and Tables

**Figure 1 ijms-23-08694-f001:**
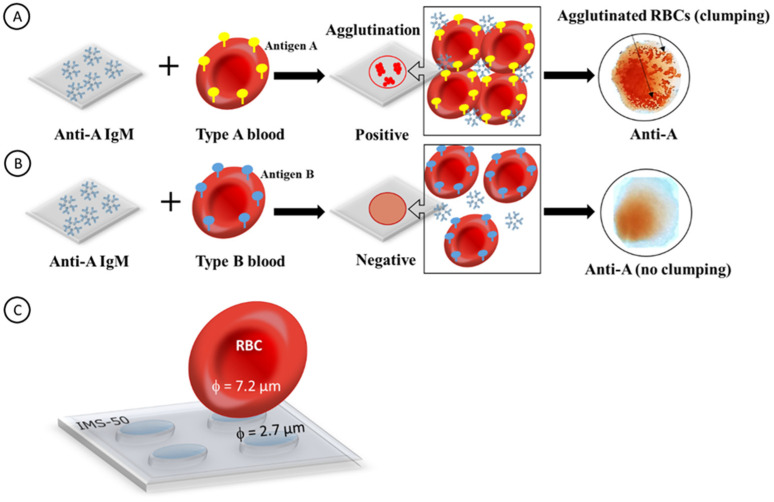
Schematic representation of the interactions of an antibody with RBCs on a paper immunostrip. (**A**) positive test: agglutination process between anti-A IgM and blood type A, (**B**) negative result: no agglutination process between anti-A IgM on paper and blood type B in the droplet, and (**C**) scheme of the RBCs deposition on the cellulose paper immunostrips IMS-50 with 2.7 µm pores.

**Figure 2 ijms-23-08694-f002:**
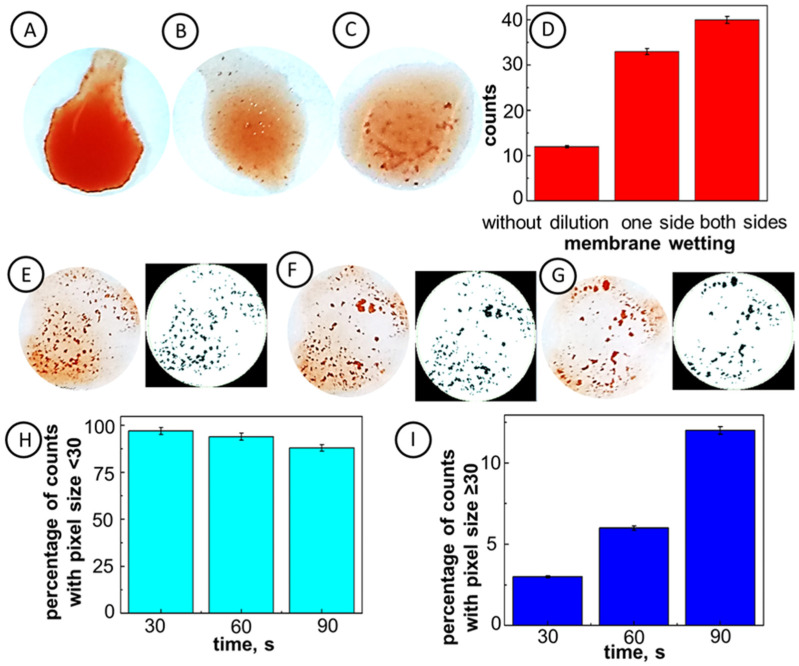
Optimization of paper wetting and interaction time for immunostrips IMS-50 for standard RBCs solution. (**A**–**D**) Effect of paper wetting with PBS: (**A**) no wetting, (**B**) unilateral (top side) wetting, (**C**) bilateral wetting, (**D**) dependence of the number of RBC aggregates on the wetting conditions for paper membrane; (**E**–**I**) optimization of the interaction time, original and rendered images for interaction times: (**E**) 30 s, (**F**) 60 s, and (**G**) 90 s; (**H**–**I**) dependence of the aggregate count percentage on interaction time for RBC aggregates with pixel size <30 (**H**) and ≥30 (**I**). All experiments were performed on immunostrips and reagents warmed to room temperature.

**Figure 3 ijms-23-08694-f003:**
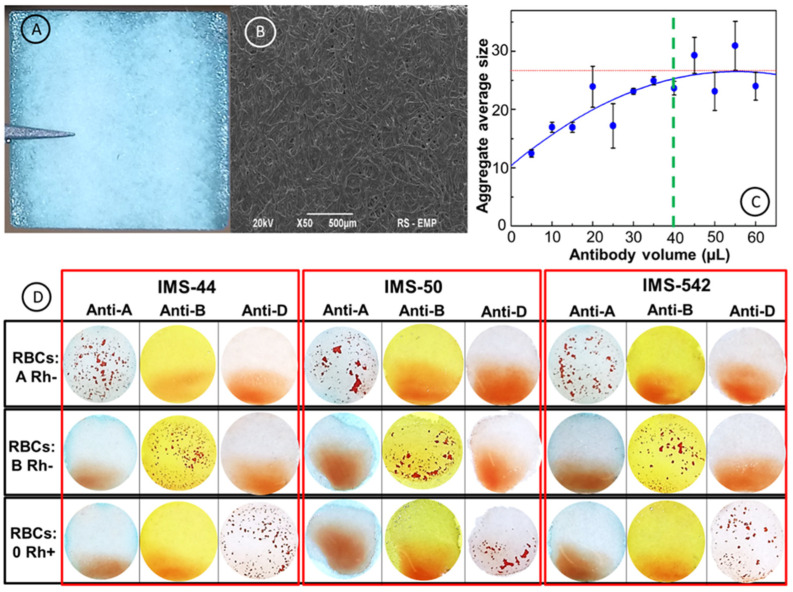
Images of paper immunostrips before and after tests with and without blood agglutination. (**A**) Optical image of a paper immunostrip; (**B**) SEM image of a paper immunostrip; (**C**) dependence of the average RBC aggregate size on antibody volume (µL); (**D**) images of immunostrips after result of interactions of standard solution of RBCs of different types: A−, B− and 0+ with antibodies (Anti-A, Anti-B, and Anti-D) immobilized on immunostrips fabricated from IMS-44 (left panel), IMS-50 (middle panel), IMS-542 (right panel) cellulose papers. All experiments were performed on immunostrips and reagents warmed to room temperature.

**Figure 4 ijms-23-08694-f004:**
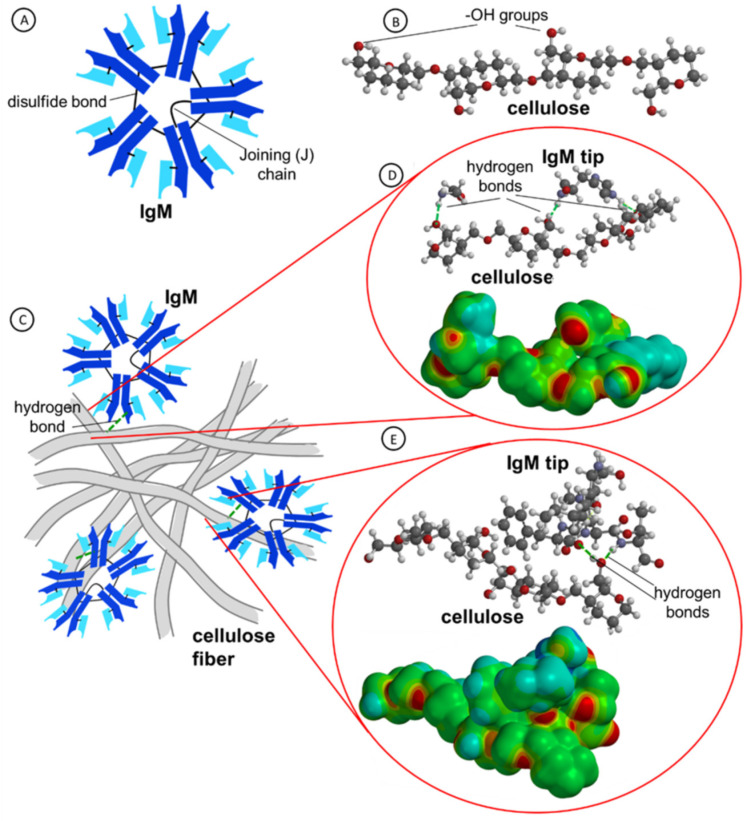
Structures and supramolecular interactions of IgM antibody with cellulose fibrils. (**A**) Schematic view of IgM structure; (**B**) structure of a cellulose chain with exposed -OH groups (red—O atoms, blue—N atoms, dark grey—C atoms, and light grey—H atoms); (**C**) cartoon of interactions of antibodies with a net of cellulose fibers; (**D**,**E**) molecular dynamics simulation of supramolecular interactions of cellulose with model amino acids (**D**) and a sequence of five amino acids (-SER-ILE-PHE-LEU-THR-) representing the IgM antibody tip: (Bottom panels) electron density surfaces of cellulose and amino acids (d = 0.002 a.u.) mapped with electrostatic potential; electrostatic potential color coding: from negative-red to positive-blue.

**Figure 5 ijms-23-08694-f005:**
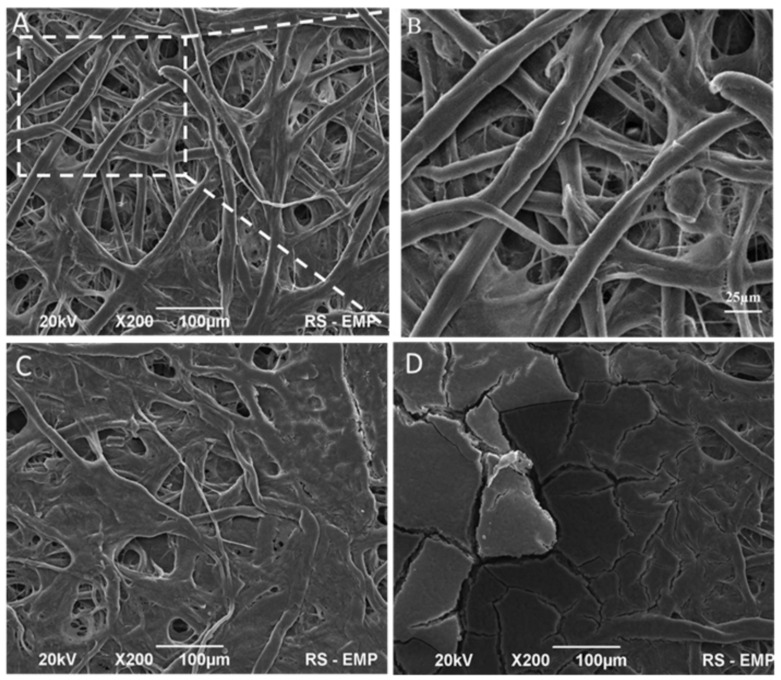
SEM images of cellulose fibers in IMS-50 immunostrip paper before blood testing (**A**,**B**) and after tests without (**C**) and with (**D**) blood agglutination. Image B is a magnification of cellulose fibers in a selected area of image A.

**Figure 6 ijms-23-08694-f006:**
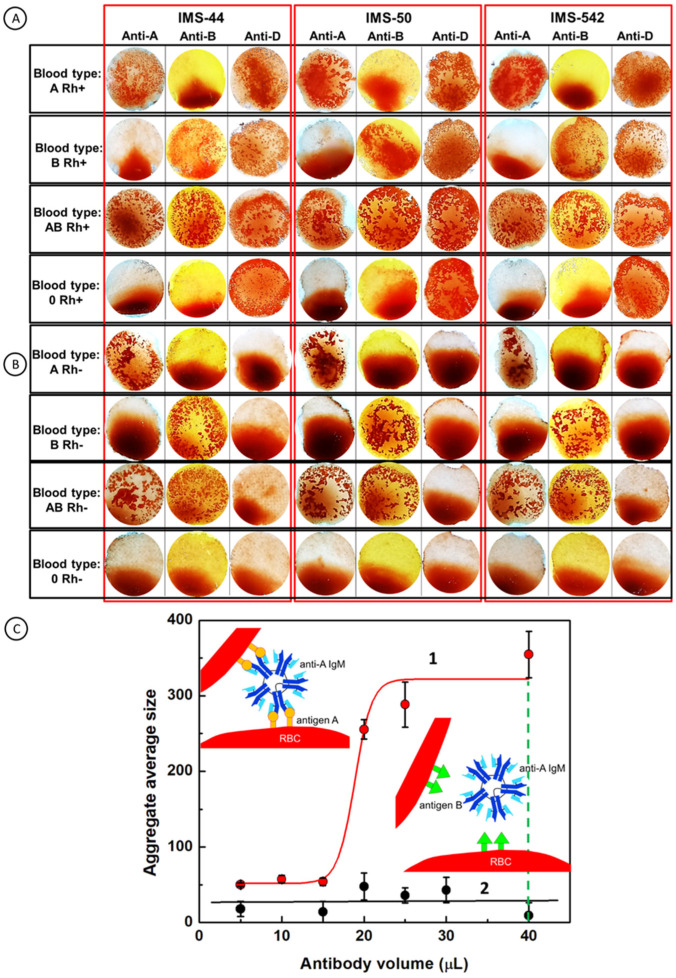
Patterns of the agglutinated RBCs formed during blood group determination on the surface of cellulose membrane immunostrips IMS-44, IMS-50, and IMS-542, as a result of interactions of blood of different types: (**A**) A, B, 0 Rh+ and (**B**) A, B, 0 Rh-, with antibodies (Anti-A, Anti-B, and Anti-D) immobilized on immunostrips, and (**C**) dependence of the average aggregate size on the antibody surface coverage for: (1) agglutination and (2) non-agglutination processes. INSET: Cartoons depicting the agglutination and non-agglutination processes. All experiments were performed on immunostrips and reagents warmed to room temperature.

**Figure 7 ijms-23-08694-f007:**
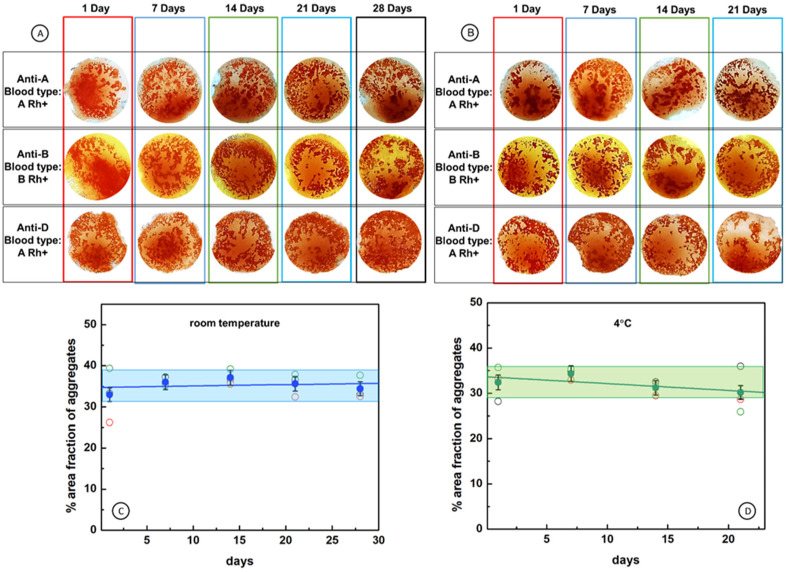
Stability of antibodies immobilized on immunostrips IMS-50 by supramolecular interactions and durability of immunostrips. (**A**,**B**) Images showing positive response of immunostrips stored at: (**A**) room temperature and (**B**) 4 °C; (**C**,**D**) durability of immunostrips stored at different temperatures: (**C**) room and (**D**) 4 °C, for different kinds of antibodies: anti-A (black open circle), anti-B (red open circle), anti-D (green open circle), mean—blue filled circle for room temperature and green filled circle for 4 °C. All experiments were performed on immunostrips and reagents warmed to room temperature.

**Figure 8 ijms-23-08694-f008:**
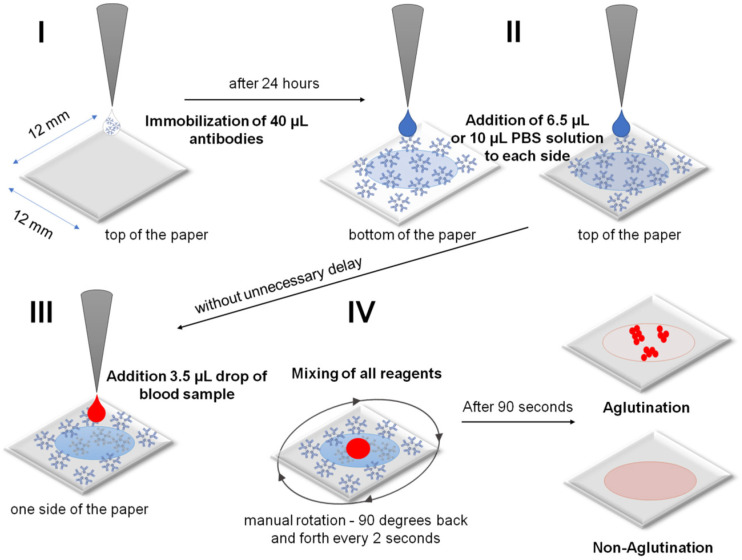
Schematic of preparation of the cellulose paper immunostrips for blood typing: (**I**) at first immobilization of antibody, (**II**) then dropping of PBS solution, (**III**) application of blood sample, and (**IV**) finally mixing of all reagents and reading out of the result.

**Table 1 ijms-23-08694-t001:** Classification of clinical blood agglutination levels for immunostrip biosensors IMS-50 for test conditions developed in this work.

Clinical Level No.	Aggregate Area, %	Agglutination Strength
-	35.0	Max
4+	28.0	Strong
3+	21.0	Heavy
2+	14.0	Moderate
1+	7.0	Mild
0	0	None

**Table 2 ijms-23-08694-t002:** Specifications of selected cellulose Whatman membranes used in the design of paper immunostrips. All membranes were derived from high quality cotton linters.

Immunostrip	Membrane Characteristic	Basis Weight [g/m^2^]	Pore Size [μm]	Thickness [μm]	Speed [s/100 mL]	Membrane Whatman Grade
IMS-50	High density	97	2.7	115	2685	50
IMS-542	Medium density	93	2.7	150	2510	542
IMS-44	Low density	80	3	180	995	44

## Data Availability

Not applicable.

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
