# Peer review of "Effective Optical Image Assessment of Cellulose Paper Immunostrips for Blood Typing"

_ijms, 2022, doi:10.3390/ijms23158694_

Round 1

Reviewer 1 Report

This paper developed a method for blood typing based on microporous cellulose paper immunostrips. The immunostrips developed are based on functionalized microporous cellulose paper matrix. Consistency of the modification should be explained in more detail. Some questions need to be answered.

 1.       Bilateral wetting cellulose paper was suggested to use in the paper. However, how to ensure the consistency of wetted area and humidity of different papers?

2.       The method of wetting is to release the antibody from the paper. How much antibody is released each time and whether it is consistent?

3.       If the antibody solution is directly added to the blood sample and then observed on paper, is it also possible? What are the advantages of this paper compared with reaction directly in antibody solution?

4.       Ims-44, ims-50, ims-542 which kind of paper works well?

5.       Antibody volume was 40ul. What is the corresponding concentration? Why use this concentration?

6.       Is the captured image area in the figure the whole area of the paper strip 12x12? If not, why not show a complete picture? If the area of blood spread is inconsistent, how to ensure the accuracy of the results?

7.       Usually, antibodies need to be preserved at 4 degrees to avoid inactivation. The immunostrip paper was stored at room temperature. Why is the activity of antibody not affected?

8.       It is suggested to add schematic diagram of modification and drying method of cellulose paper.

Reviewer 2 Report

The introduction section should be improved. The current state of the blood group immunostrip research field should be reviewed carefully (in terms of cellulose membrane fabrication, membrane coating with antibodies, loading volume, and antibodies concentration), and key publications should be cited. Please highlight the controversial and diverging development process of the immunostrip. The last paragraph of the introduction summarizes part of the research optimization outcomes, which should be part of the method and results. This paragraph also demonstrated the research results, which should be part of the discussion and research conclusion.

The method and result section should be revised and improved to show the research development process. The current organization of this section lack coherence and does not show the actual developmental process of the technique and the procedure of the immunstrip model. It is recommended to modify it as the following: proposed model principal, membrane fabrication (including density, weight, pore size, thickness, and membrane fabrication shape, size, speed and grade), antibodies conjugation (including antibody type, concentration, loading, timing, immobilization and stability), blood typing procedure (including immunostrip conditioning volume, agglutination count and strength, experimental timing).

Authors must indicate the RBCs counts and volumes (dilution factor if applicable) used in all experiments to ensure constancy and accuracy. Authors have to include more results for the negative control (uncoated strip) in all experiments for confirmation that the haemagglutination accrued due to the antigen-antibody interaction.

According to figure 2 (H and I), the percentage of small aggregates decreases with time, and large aggregates increase with time. This is since Micro haemagglutination formation is influenced by time, temperature, microwell shape and size, antibody types, infinity and titration. None of these factors was evaluated and optimized and was not standardized in the method section or the results. These findings require more clarification as they contradict the authors' claim. In paragraph 5, section 2.2 of the results (the Optimization of experimental conditions) authors discussed the contributing factor for agglutination. This paragraph should be part of the discussion as no optimization results showed.

In figure 3, the author indicated the change in aggregate size according to the antibody volume (ul, according to figure C). However, in the figure legend author indicated that they evaluated the aggregate size of the antibody surface coverage. Clarification is needed, and an accurate measuring unit show is used. It is recommended to show the actual antibody concentration if possible.

In Figure 4, patterns of agglutination in different experimental settings, please confirm if the authors are using the volume of the antibody (ul) of the surface coverage (um). Additionally, figure C shows that small RBS aggregates (>100u) are also formed in their model for nonspecific antibody-antigen interaction. This point requires more explanation and clarification. Are they due to the strip fabrication or spore size, which can produce false positive results?

The authors indicated the impact strip durability after storage at room or 4-degree temperatures. However, the current model lacks temperature evaluation for the experimental setting. This step is essential in the blood bank experimental setting at the temperature influencing the antigen-antibody interaction and aggregate formation.

The current results were not showing the optimal cellulose strip specifications and how that impacts aggregate formation and results reading. It is recommended to reorganize the method and results section to appropriately show the importance of the proposed procedure and the optimization protocol.

The result section included some experimental outcome that was not described in the method section. Additionally, some paragraphs discussed the experimental outcomes that should be part of the optimization evaluation of the proposed procedure. Results subheadings are not matching with the results shown in the manuscript.

The authors did not discuss the results appropriately and how they can be interpreted from the perspective of previous studies and of the working hypotheses. The authors discussed the factors contributing to the outcome of their work, which should be part of their optimization protocol.

Some of the references were outdated; please use updated references.  

Authors are recommended to add more supplement material to show their optimization protocol and its outcome due to the limitation in pictures and figure counts.

Reviewer 3 Report

This is a useful and interesting study, and I would like to recommend this manuscript for publication. I just have one small question:

It should be ABO or AB0? Also the 0+,0-, are they O+, O-?

Author Response

We would like to thank very much the Reviewer for his valuable comments on our article. We believe that the problem raised by the reviewer is only a problem with the font used. 

Round 2

Reviewer 1 Report

Accept in present form

Reviewer 2 Report

all comments were addressed 

English require editing especially for the new paragraphs added after modification